# Apollo: Adaptive Polar Lattice-Based Local Obstacle Avoidance and Motion Planning for Automated Vehicles

**DOI:** 10.3390/s23041813

**Published:** 2023-02-06

**Authors:** Yiqun Li, Zong Chen, Tao Wang, Xiangrui Zeng, Zhouping Yin

**Affiliations:** 1School of Mechanical Science and Engineering, Huazhong University of Science and Technology, Wuhan 430074, China; 2School of Mechanical Engineering and Automation, Harbin Institute of Technology, Shenzhen 518071, China

**Keywords:** automated vehicle, motion planning, adaptive polar lattice, obstacle avoidance

## Abstract

The motion planning module is the core module of the automated vehicle software system, which plays a key role in connecting its preceding element, i.e., the sensing module, and its following element, i.e., the control module. The design of an adaptive polar lattice-based local obstacle avoidance (APOLLO) algorithm proposed in this paper takes full account of the characteristics of the vehicle’s sensing and control systems. The core of our approach mainly consists of three phases, i.e., the adaptive polar lattice-based local search space design, the collision-free path generation and the path smoothing. By adjusting a few parameters, the algorithm can be adapted to different driving environments and different kinds of vehicle chassis. Simulations show that the proposed method owns strong environmental adaptability and low computation complexity.

## 1. Introduction

The paper aims to give an efficient local motion planning method for automated driving in complex environments. The planning module, which links the system perception module and the control module, plays a special and momentous role in the automated driving system. It needs to reasonably organize and formulate the sensing, positioning and routing information while considering the status of the vehicle. An eximious motion planner of an automated vehicle needs to consider the safety, feasibility, efficiency, environmental adaptability and real-time performance of the algorithm, especially in dynamic driving scenarios with humans.

### 1.1. Related Work

A hierarchical framework is typically adopted, in which the motion planning task is decomposed into the global path planning and the local path planning problems. The global path planning or deliberative path planning is typically map-based and often computed offline, while the local path planning or reactive planning is sensor-based and conducted online. Global planning algorithms for completely known topological or occupancy grid maps, such as A*, D*, jump point search (JPS), probabilistic roadmap (PRM), rapidly exploring random tree (RRT), Voronoi graph and their variants [1,2,3,4,5], are well studied. In practice, global path planning is often problematic due to the inaccuracy or unavailability of the global world model. The local path planning method is mainly used for real-time collision avoidance and global path tracking in unknown or semi-unknown environments via the onboard sensing information. Many local path planning algorithms are prone to falling into local minima or exhibiting oscillating behaviors. This paper proposes an adaptive polar lattice-based local obstacle avoidance and motion planning method (APOLLO) which is effective and computationally efficient in dynamic driving scenarios with humans. Without loss of generality, the vehicle’s sensing area is assumed to be fan-shaped. APOLLO is conducted on the scalable polar grids around the vehicle with elaborately designed probabilistic representations. A coarse collision-free path can then be generated by connecting grid nodes according to some optimization criteria. An efficient strategy is also proposed to adjust the selected path to a kinodynamic, plausible smooth trajectory. A local planner should track the global path while avoiding dynamic obstacles as well as taking into consideration the vehicle constraints. Several methods have been successfully used in some distinct driving scenarios, such as potential field methods [6], the dynamic windows approach (DWA) [7], the time-elastic band (TEB) [8,9,10], a state lattice [11,12,13], a numerical optimization approach [14,15,16,17], machine learning methods [18,19], etc. Paths planned by the potential field methods are generally smooth and safe, but methods of this kind have the problem of local optima, and it is difficult for the vehicle to approach the target point by using such methods when there are obstacles near the target point. The DWA algorithm consists of three main processes, i.e., the velocity sampling, the obstacle detection and the trajectory evaluation. This algorithm selects the path with the highest score and sends the associated speed to the vehicle, and it is also susceptible to local minima [20]. The TEB algorithm is formulated as a multi-objective nonlinear optimization problem with the initial trajectory generated by a global path planner, which can take into consideration the geometric and kinematic/dynamic constraints of the vehicle. Thanks to efficient optimization techniques, such as the “g2o” framework, the TEB algorithm can be implemented robustly and efficiently in a simple environment with relatively inexpensive hardware. However, this kind of algorithm may let the vehicle wander around the channel entrance when facing a narrow channel environment (Figure 1a). State lattice approaches, such as FALCO [11] (Figure 1b), which are based on pre-computed motion primitives and online path selection, can effectively reduce the computational complexity while needing relatively larger storage spaces, and they may produce oscillatory behavior in front of long obstructions. The receding horizon optimization (RHO) and model predictive control (MPC) methods transform the local trajectory generation problem into the numerical optimization problem, which can easily take into account the environment and vehicle constraints in the model and keep the ride smooth [21]. However, this kind of approach requires a deliberate balance between the model complexity and the computational complexity.

### 1.2. Contribution

A probability-based method to construct an adaptive local polar lattice search space with obstacles is proposed. Unlike an equidistant lattice, the proposed adaptive polar lattice based on sensor field of view (FOV) is closer to human search habits. The concise collision-free path generation and smoothing method has high computational efficiency. Compared with some state-of-the-art methods, the proposed obstacle avoidance and motion planning method shows stronger environmental adaptability and is effective in dynamic driving scenarios.

## 2. Local Search Space Design

An occupancy grid map is one of the most commonly used scene representation methods for automated vehicles. According to the size of the 2D grids, it can be divided into uniform grids, non-uniform grids and adaptive size grids. In practical applications, adjusting the local resolution of the grids adaptively according to the complexity of the environment and driving speed of the vehicle will be very helpful for motion planning. Despite the computational advantages, the variable-size grids can implicitly take the uncertainty of local sensing information into account. Apart from Cartesian occupancy grid maps, polar coordinate-based grid maps are also used especially in outdoor environments, having the advantages of easy adjusting of the angular resolution and integration of obstacles perceived by sensors, such as cameras and lidars [22].

### 2.1. Adaptive Polar Lattice

As shown in Figure 2, the angle range β and radius *L* of the automated vehicle *M* are set adaptively according to the sensor horizontal FOV, the sensing range and the environment. Rays are built to divide the circumferential local search space, and the circular arcs with the automated vehicle as the center divide the radial direction of the space. The rays and circular arcs divide the local search area into m×n polar lattices with different sizes, where m denotes the amount of radial dispersion and n the amount of angle dispersion. Each polar lattice in Figure 2 can be assigned a probability. The magnitude of each grid’s probability value depends on the likelihood of the vehicle traversing that grid. Then, the overall probability distribution of all the polar lattices can be stored in an m×n matrix,
$$\Omega =\left[\omega_{ij}\right],\;i=1,\cdots ,m;j=1,\cdots ,n.$$

Suppose the size of the angle and radial interval, i.e., $\delta_{a},\delta_{r}$, are constant, and the probability of any point $\left(x_{k},y_{k}\right)$ in the local search space is assigned to be $\omega_{ij}$. Here, the grid index i,j corresponding to the point $\left(x_{k},y_{k}\right)$ can be calculated by Equations (1) and (2), when $\delta_{a},\delta_{r}$ are constant.
(1)$$i=\left\lfloor \frac{\sqrt{x_{k}^{2}+y_{k}^{2}}}{\delta_{r}}\right\rfloor ,$$
(2)$$j=\left\lfloor \frac{\arctan\left(y_{k}/x_{k}\right)}{\delta_{a}}\right\rfloor .$$

Here, ⌊⋅⌋ means rounding down. In real applications, $\delta_{a},\delta_{r}$ are tuned adaptively according to the geometric and physical parameters of the vehicle and the complexity of its working environment. Without a loss of generality, $\delta_{a}$ is set to be constant in the following discussion for simplicity. Considering that the motion planning of a vehicle should be more conscientious for the area closer to itself than the area far away, $\delta_{r}$ can be set as an arithmetic progression, a geometric progression or a logarithmic function. Similar to (1) and (2), the index i,j of any point in the search area can also be easily calculated for the case that $\delta_{r}$ is an arithmetic or geometric progression. A hash table is used to store the mapping between each point’s coordinates and the corresponding index of Ω. The complexity of the indexing time is O(1) (see Figure 2). Compared with the equidistant grid, the proper selection of the form and value of $\delta_{a},\delta_{r}$ can reduce the computational complexity of the motion planning algorithm while ensuring the safety of the vehicle. We do not claim that an arithmetic progression or a geometric progression is the best choice for $\delta_{r}$. It needs to be determined according to the actual driving conditions.

### 2.2. Local Probability Distribution

An automated vehicle with nonholonomic constraints cannot move laterally. In order to make the vehicle ride smoother and reduce frequent turns, the probability should be set higher in the vehicle’s heading direction. As show in Figure 3, for a given initial state $x_{0}$, the vehicle can take different paths to the sensor frontier F. The 2D path first splits in 37 directions, and each path splits in another 12 directions. For each path generated as a cubic spline curve, the resulting paths group is symmetric. As the end points of the local search space are reachable, the probabilities at the edge of the search area should be approximately consistent.

Based on the above considerations, the initial probability distribution function of the local search space without obstacles and with a goal point is designed as (3), which is similar to a two-dimensional Gaussian distribution.
(3)$$f_{X}\left(r,a\right)=\eta \frac{1}{\sqrt{{\left(2\pi \right)}^{2}\left|\Sigma \right|}}e^{-\frac{1}{2}{\left(X-\mu \right)}^{T}\Sigma^{-1}\left(X-\mu \right)}$$
where
$$X={\left(r,a\right)}^{T},\;\mu ={\left(\mu_{r},\mu_{a}\right)}^{T},\;\Sigma =\left|\begin{array}{cc} \sigma_{r}^{2} & \rho \sigma_{r}\sigma_{a}\\ \rho \sigma_{r}\sigma_{a} & \sigma_{r}^{2} \end{array}\right|.$$

Here, r represents the distance between the point in the search range and the vehicle, and a represents the polar angle of the proposed point in the vehicle’s coordinate system. μ is the expectation of r and a. Σ represents the covariance matrix. η and ρ are constants. Figure 4 illustrates the initial probability distribution of a 180° fan search space.

In order to make the vehicle move towards the target point, it is necessary to set high probabilities along the direction of the target point. As shown in Figure 5, we denote the distance and azimuth of the target point to be $r^{*}$ and $a^{*}$, respectively.

Then, the probability distribution between the vehicle and target point is set as:(4)$$g\left(r,a\right)=\frac{\zeta r}{r^{*}}\frac{1}{\sqrt{2\pi \left|\sigma \right|}}e^{-\frac{1}{2}{\left(a-a^{*}\right)}^{T}\sigma^{-1}(a-a^{*})}$$
where $r\in [0,r^{*}]$ and $a\in [a^{*}-\Delta a,a^{*}+\Delta a]$. For a distinct distance r from the original point, Formula (4) degenerates to a Gaussian distribution with respect to the angle a. Here, $a^{*}$ and σ are respectively the mean and variance of the proposed Gaussian distribution.ζ is a constant, and Δa is an adjustable parameter.

### 2.3. Obstacle Avoidance

It is always expected that the planned path of the vehicle will be as far away from obstacles as possible, so as to ensure safety. However, a too-conservative path planning strategy will inevitably reduce the planning efficiency and increase the length of the resulting path. As shown in Figure 6, the vicinity region of the obstacle point cloud $p=\left(x_{k},y_{k}\right)$, k=1,2,⋯, is divided into two areas, i.e., the strictly prohibited area and the expansion area. Here, $r_{0}$ denotes the strictly prohibited region, which is related to the shape and size of the vehicle.

The probability distribution around the obstacles is updated according to the following piecewise function:(5)$$\Omega \left(i,j\right)=\left\{ \begin{array}{cc} -\gamma , & \left\|p_{0}-p_{i,j}\right\|<r_{0}\;\\ -\frac{\gamma }{\left\|p_{0}-p_{i,j}\right\|+\varepsilon }, & r_{0}\leq \left\|p_{0}-p_{i,j}\right\|\leq r_{s} \end{array}\right.$$

Here, $p_{0}$ represents the coordinates of the obstacle point cloud. $r_{s}$ denotes the radius of the expanded obstacle point cloud, and γ,ε are adjustable parameters. Figure 7 illustrates the probability distribution of the local search space with two static obstacles at coordinates (−1.0, 1.5) and (0.0, 0.7).

## 3. Collision-Free Path Generation

Based on the above local search space design, the problem of collision-free path planning can be transformed into the problem of node selection, i.e., the nodes are selected and connected one by one along the arcs that gradually expand from near to far in the search area and in the meantime ensure that the connecting lines do not intersect with obstacles (see Figure 8).

As shown in Figure 9, the azimuth angle of the goal point in the sector search area is first calculated. For a goal point inside the local search space, the node closest to the goal is chosen as the target. For the case that the goal point is outside the local search space, the center point of the vehicle is connected to the goal point, and the intersection point with the boundary of the search area is denoted as *E*. If *E* is collision-free, then its closest node is selected as the temporary target point. Otherwise, the nodes around *E* are selected as potential targets. The path weight of the vehicle to each potential target is calculated, and the path with the largest value is selected.

The specific path generation procedures are given in Algorithm 1.
**Algorithm 1** Collision-free path planning**Initialize** weight matrix Ωselect the node E∈S closest to the goal and calculate its corresponding index (i,j) in matrix Ω**If**Ω(i,j)<0 **then   //** indicates the lattice with index (i,j) is occupied  find neighbor node $E^{b}$ of E, b=1,2,⋯ generate corresponding paths $p^{b}$ calculate total weight $\omega^{b}=\sum_{i=1}^{m}\omega_{i}$ of each path select path $p^{*}=\{p^{b}|b=\arg\underset{b}{\max}\omega_{b}\}$ **return**
$p^{b}$**else** generate path $p^{0}$ from the vehicle to node *E* **if**
$p^{0}!=\mathrm{NULL}$ **then**  return $p^{0}$ **else**   repeat steps 4–8  **end if****end if**

## 4. Path Smoothing

In order to make the tracking process of the vehicle smoother and more stable, it is necessary to convert the polyline into a kinodynamic-friendly trajectory, especially at sharp turning points. The path transferring process should also be collision-free and can be adjusted automatically according to the speed of the vehicle.

### 4.1. Fast Point Sequence Generation

As shown in Figure 10, the unit heading vector of the vehicle at the position $A_{i}$ is denoted by $v^{i},i=0,1,\cdots$. Point $B_{i}$ is on the vehicle’s current heading vector, and the distance between $A_{i}$ and $B_{i}$ is set to be $l_{i}$. With $B_{i}$ as the center point, find point $C_{i}$ on the polyline path so that $\left\|B_{i}C_{i}\right\|=\kappa_{i}$. Find point $A_{i+1}$ on vector $B_{i}C_{i}$ so that $\left\|B_{i}A_{i+1}\right\|={l^{\prime }}_{i}$. Denote as $v_{i+1}$ the vector starting from $A_{i+1}$ and going in the direction of the extension line $A_{i}A_{i+1}$ (see the green vector in Figure 10). The above process is repeated until the vehicle reaches the vicinity of the target point, and the point sequence $A_{i},i=0,1,\cdots$ generated in the process is smooth.

The point sequence can be kept collision-free by adjusting the parameters $\kappa ,l,l^{\prime }$. Algorithm 2 gives the specific path smoothing procedures.
**Algorithm 2** Path smoothing**Initialize** weight matrix Ω and $\kappa ,l,l^{\prime }$  // $\kappa ,l,l^{\prime }$ are adjustable parameters**while** $\left|A_{i}E\right|>\varepsilon$ **do   //**ε is the margin   $B_{i}=A_{i}+l_{i}\cdot v^{i}$   // $v^{i}\int vccc$ is the orientation of the vehicle   find point $C_{i}$ so that $\left|B_{i}C_{i}\right|=\kappa$   choose test points $D_{1},D_{2},\cdots ,D_{s}$ on $B_{i}C_{i}$   **for** k=1:s **do**    calculate index (i,j) of point $D_{k}$ in Ω    **if**
Ω(i,j)<0 **then**      κ=κ−Δκ      back to step 3     **end if**
  **end for**   $A_{i+1}=B_{i}+{l^{\prime }}_{i}\cdot \vec{B_{i}C_{i}}/\kappa$  calculate angular velocity between $A_{i}$ and $A_{i+1}$**end while**

As shown in Figure 10, a sequence of points $\left\{A_{0},A_{1},\cdots \right\}$ with directions can be generated by Algorithm 2. Figure 11 shows the path smoothing results of differently shaped polyline paths with different parameters $\kappa ,l,l^{\prime }$.

Figure 12 shows the path smoothing results of Algorithm 2 in a driving scenario with obstacles.

### 4.2. Speed Optimization

The smooth path generated can be followed by path tracking methods. Usually, the control module should be fed with the linear velocity and angular velocity at a given moment. Several approaches can be applied to transfer the discrete path into v,ω with limited motor torques, such as multi-objective optimization [8,9], polynomial interpolation and quadratic programming [14]. The optimization can also be performed on the *SE*(2) manifold, and then the resulting pose can be directly tracked by geometric control methods [10,23]. In this paper, it is assumed that the velocity between two adjacent configurations
g0=(x0,y0,θ0), ge=(xe,ye,θe)∈ℝ2⋉S1≃SE(2)
is constant and that the direction of it has been obtained in the path smoothing process. Here, (x0,y0),(xe,ye) denote the position of the vehicle, and θ0,θe denote the orientation of the vehicle. Divide the time interval T=[t0,te] between g0 and ge into *N* equal subintervals with length δt.

As shown in Figure 13, the parameters selected in Algorithm 2, such as $l_{i},l_{i}{}^{\prime }$, are closely related to the kinematics of the vehicle. For example, if $l_{i}=l_{i}{}^{\prime }$, the path between $A_{i}$ and $A_{i+1}$ is a circular arc. According to the geometric relation, the formula for solving the velocity can be obtained as:(6)$$v=\frac{\sqrt{{\left(x_{e}-x_{0}\right)}^{2}+{\left(y_{e}-y_{0}\right)}^{2}}}{2\sin\frac{\theta_{e}-\theta_{0}}{2}}\cdot \frac{\theta_{e}-\theta_{0}}{T}$$

Then, the angular velocity $\omega_{i},i=1,\cdots ,N$ between the configurations $g_{0}≜\left(p_{0},\theta_{0}\right)$ and $g_{e}≜\left(p_{e},\theta_{e}\right)$ can be obtained by the following optimization model:(7)$$\begin{array}{l} \underset{\omega_{i}}{\min}\;J_{\theta }+J_{p}\\ s.t.\;\omega_{\min}\leq \omega_{i}\leq \omega_{\max},\;i=1,\cdots ,N\\ \;\;\;\Delta \omega_{\min}\leq \omega_{i+1}-\omega_{i}\leq \Delta \omega_{\max},\;i=1,\cdots ,N-1 \end{array}$$
where
(8)$$\begin{array}{l} J_{\theta }=\alpha_{1}{\left\|\left(\theta_{0}+\sum_{i=1}^{N}\omega_{i}\delta t\right)-\theta_{e}\right\|}_{2}\\ J_{p}=\alpha_{2}{\left\|\left(p_{0}+\sum_{i=1}^{N}v_{i}\delta t\right)-p_{e}\right\|}_{2} \end{array}$$

Here, $\alpha_{1},\alpha_{2}$ are weights corresponding to the terminal angle and position error. The vector v in (8) consists of two elements, i.e., $v={\left[v_{x},v_{y}\right]}^{T}$. The above objective function can also be represented by the distance metric of *SE*(*2*), which is $\left\|\log\left(g_{0}{}^{-1}g_{e}\right)\right\|$(here we abuse the notations $g_{0},g_{e}$ to represent group elements in *SE*(*2*)). For *N* = 3, the above optimization formulation can be relaxed to the following algebraic equations:(9)$$\begin{array}{l} \theta_{e}=\theta_{0}+\left(\Delta \theta_{1}+\Delta \theta_{2}+\Delta \theta_{3}\right)\\ x_{e}=x_{0}+\Delta s\left[\cos\left(\theta_{0}+\theta_{1}\right)+\cos\left(\theta_{0}+\sum_{i=1}^{2}\Delta \theta_{i}\right)+\cos\left(\theta_{0}+\sum_{i=1}^{3}\Delta \theta_{i}\right)\right]\\ y_{e}=y_{0}+\Delta s\left[\sin\left(\theta_{0}+\theta_{1}\right)+\sin\left(\theta_{0}+\sum_{i=1}^{2}\Delta \theta_{i}\right)+\sin\left(\theta_{0}+\sum_{i=1}^{3}\Delta \theta_{i}\right)\right] \end{array}$$
where $\Delta \theta_{i}=\omega_{i}\delta t$ and Δs=vδt. It is obvious that Equation (9) has a unique analytical solution. As $\theta_{i}$ are often very small, especially with high control frequencies, in practice, the trigonometric functions in (9) can be approximated by the first-order Taylor expansion. Figure 14 and Figure 15 show the linear and angular velocity, i.e., v,ω, generated in the long wall driving scenario.

It should be pointed out that the velocity and time can also be regarded as the decision variables in the optimization model, but this will consume much more time for solving the resulting optimization problem. Based on the consideration of real-time performance, the model is simplified, and the resulting model is verified to be effective and efficient. Due to the extremely low computational complexity of the proposed path planning and smoothing algorithms, the method can be effectively applied to human-involved dynamic driving scenarios without predicting the motion of the humans.

## 5. Simulations

The performance of the proposed method is evaluated in different simulated scenarios (see Figure 16): the forest scenario, the narrow channel scenario, the long wall scenario and the human-involved dynamic driving scenario. The algorithms are implemented in C++ and are executed on a laptop running Ubuntu 16.04 with an Intel i5-10200H CPU and 16 GB of RAM.

The TEB algorithm is also tested in the narrow channel scenario. Figure 17 shows the simulations of the TEB algorithm with different maximum global looking-ahead distances and the performance of the APOLLO algorithm in the narrow channel scenario. It can be seen that the vehicle is wandering around the gate and cannot follow the global path, while the APOLLO algorithm allows the vehicle to pass smoothly through the narrow channel of the same size. For the long wall scenario, the FALCO algorithm exhibits oscillatory behavior (Figure 1b), while the APOLLO algorithm can choose a relatively better direction without hesitation (Figure 16c).

Figure 18 records the CPU times of various planning algorithms in the long wall simulation environment. As we can see, APOLLO has the highest time efficiency and needs less than 2 ms on average to complete the local planning task. Compared with the state lattice-based FALCO local planner, the APOLLO planner does not need to pre-compute the path group for a distinct vehicle and only needs to change few parameters to adjust the size of the desired sector region, which makes APOLLO well adapted to different driving speeds of the automated vehicle.

Figure 19 shows the trajectory planning process of the automated vehicle in the human involved dynamic driving scenario. Figure 20 and Figure 21 show the linear and angular velocity, i.e., v,ω, generated in this dynamic driving scenario.

Owning to its strong environmental adaptability and low computational complexity, the proposed APOLLO algorithm can also be used by the automated vehicle in a fleet to generate local collision-free trajectories. This algorithm can be applied in dynamic indoor environments, such as smart factories, shopping malls or airport stations, as well as high-speed outdoor transportation environments.

## 6. Conclusions

This paper presents APOLLO, a fast local smooth path generator for agile drives in unknown dynamic environments. The key advantage of the proposed local planner lies in its low computational complexity and its adaptivity. Therefore, this method can deal with human-involved dynamic driving scenarios and high-speed driving scenarios. Some of the variable parameters in the proposed algorithms can be adjusted manually to adapt to different driving scenarios.

## Figures and Tables

**Figure 1 sensors-23-01813-f001:**
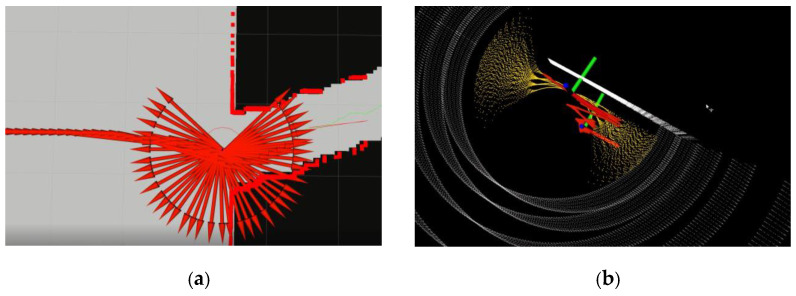
Oscillatory behavior of TEB and FALCO. (**a**) TEB in narrow channel; (**b**) FALCO in front of long obstruction.

**Figure 2 sensors-23-01813-f002:**
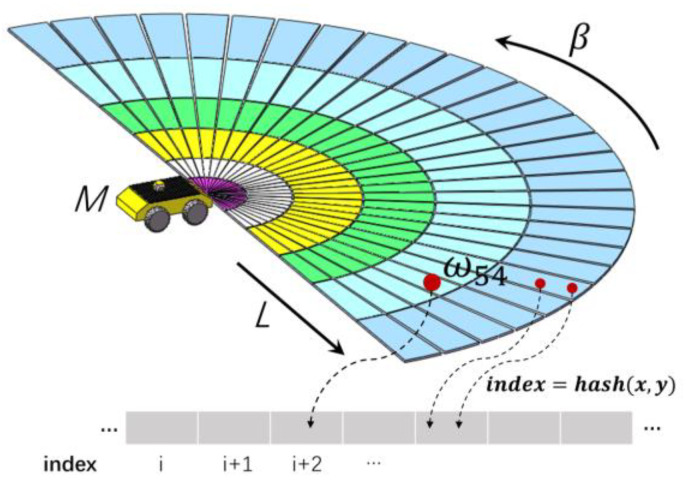
Local search space partition.

**Figure 3 sensors-23-01813-f003:**
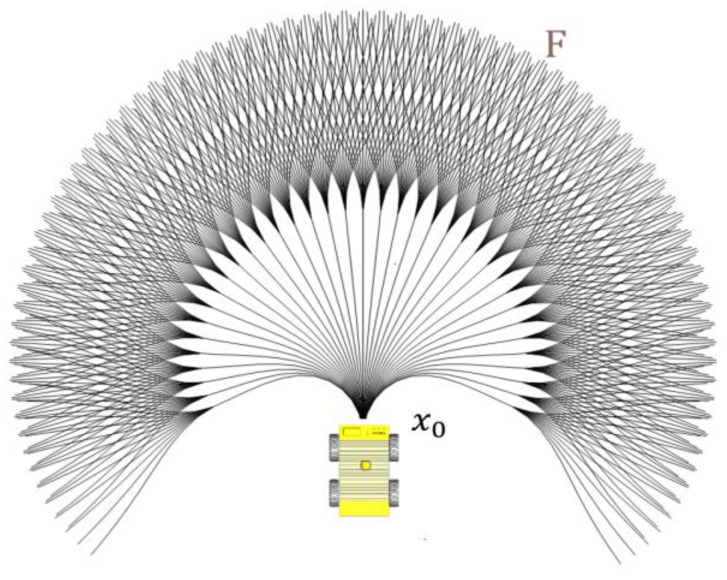
Illustration of the state lattices of the nonholonomic robot.

**Figure 4 sensors-23-01813-f004:**
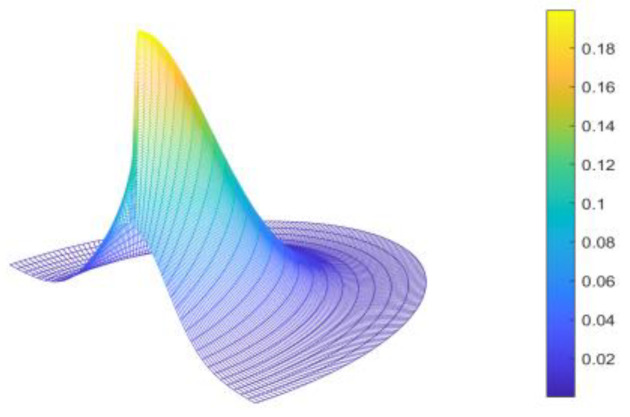
3D illustration of the initial probability distribution.

**Figure 5 sensors-23-01813-f005:**
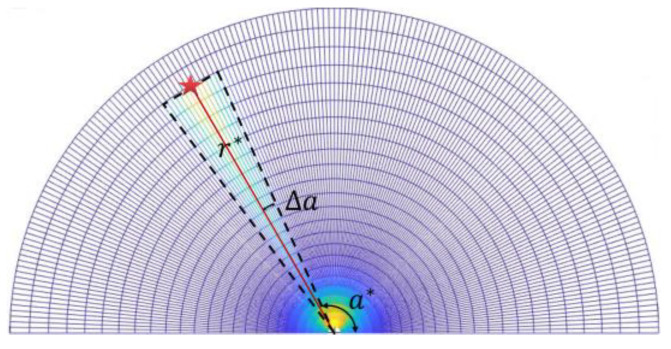
Probability distribution along the direction of the target point in a 180° fan search space.

**Figure 6 sensors-23-01813-f006:**
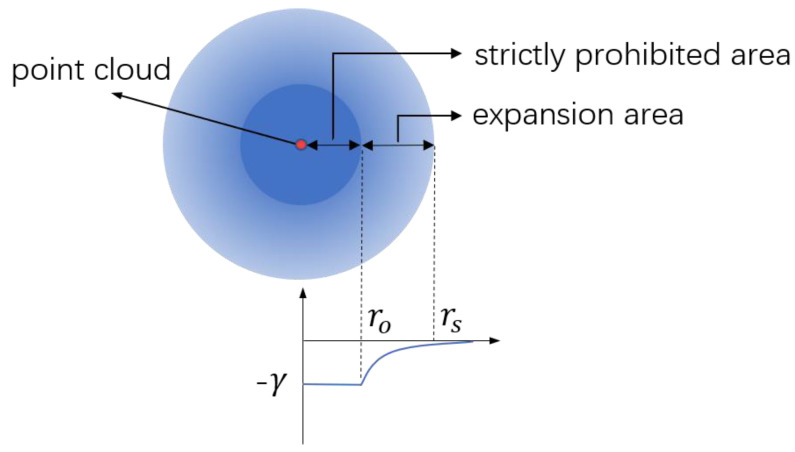
Expansion of an obstacle point cloud.

**Figure 7 sensors-23-01813-f007:**
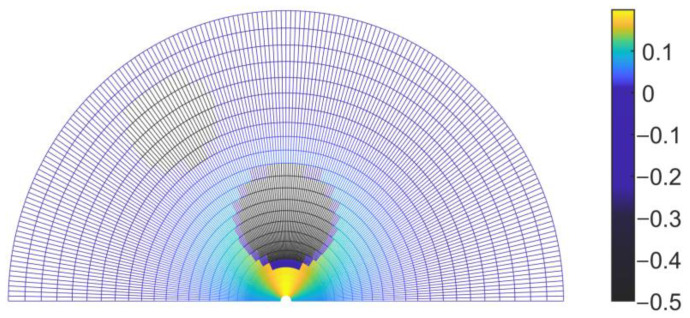
Probability distribution of the local search space with obstacles in polar coordinates.

**Figure 8 sensors-23-01813-f008:**
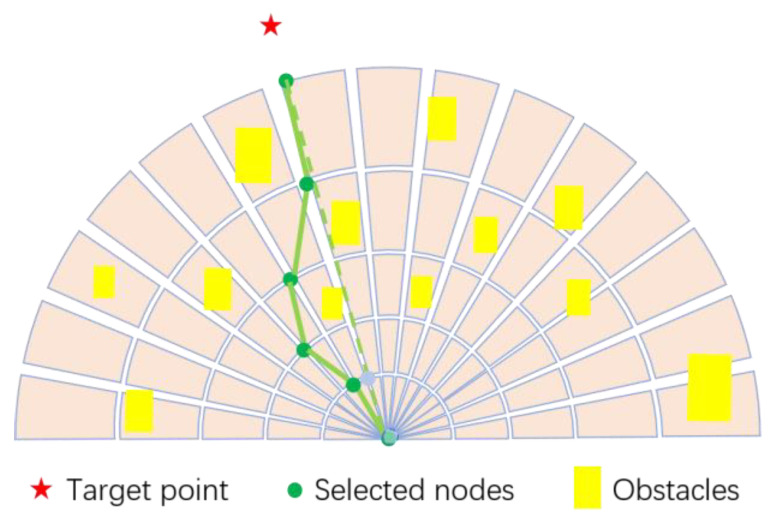
Node selection and connection.

**Figure 9 sensors-23-01813-f009:**
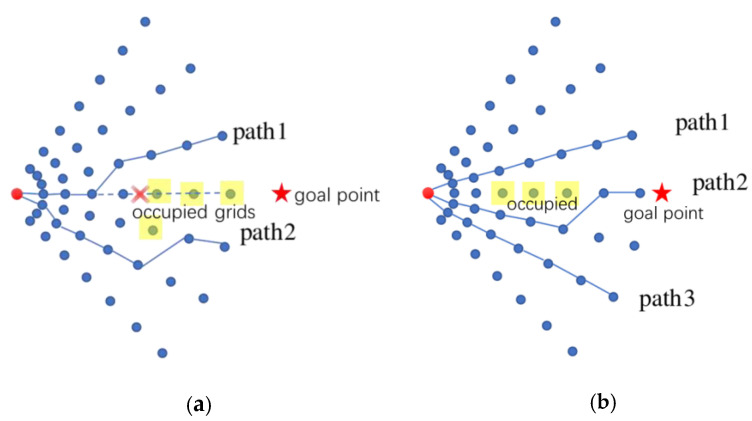
Illustration of path planning when (**a**) target is occupied; (**b**) target is collision-free.

**Figure 10 sensors-23-01813-f010:**
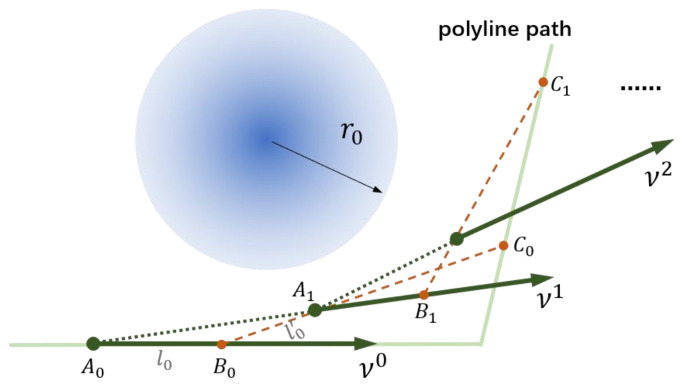
Illustration of path smoothing procedures.

**Figure 11 sensors-23-01813-f011:**
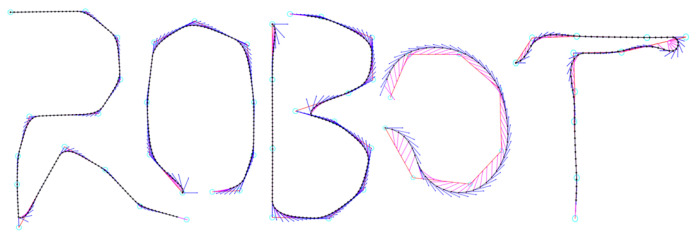
Path smoothing of differently shaped polyline paths.

**Figure 12 sensors-23-01813-f012:**
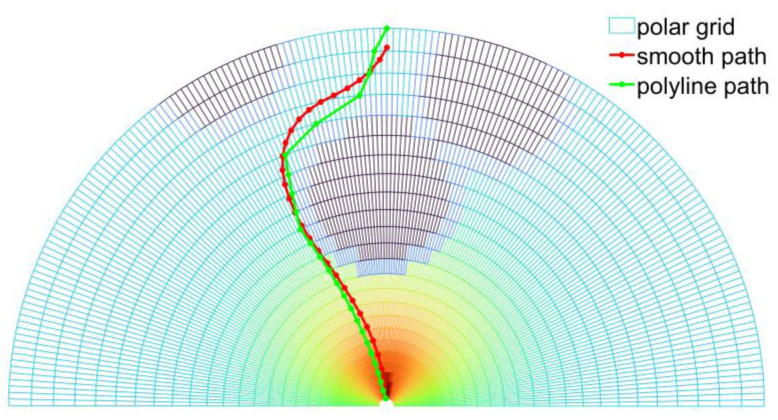
Illustration of the generated smooth, collision-free paths.

**Figure 13 sensors-23-01813-f013:**
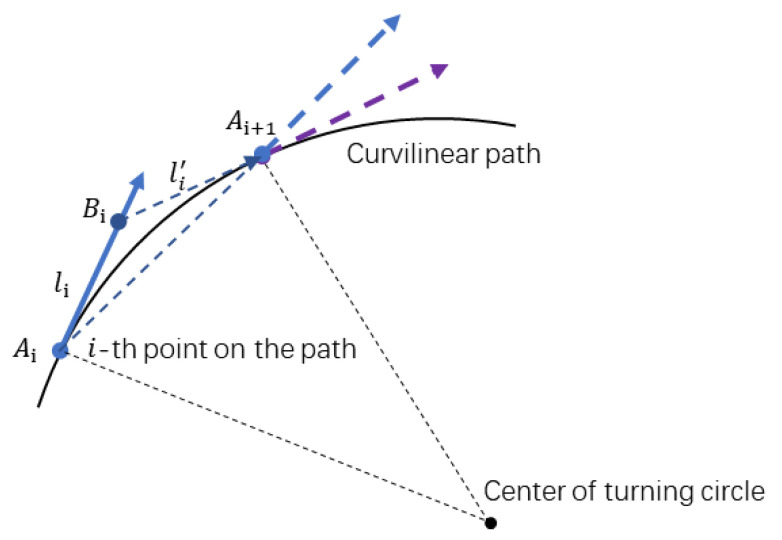
Illustration of geometric relations between two poses.

**Figure 14 sensors-23-01813-f014:**
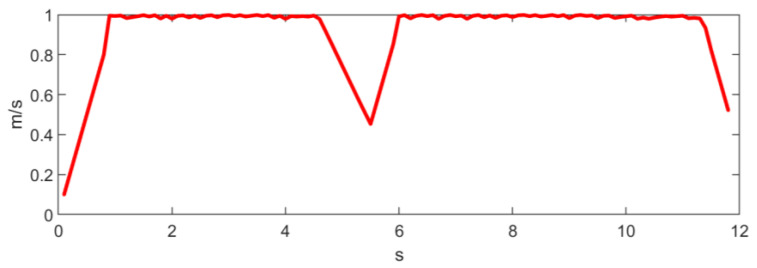
Linear velocity generated in the long wall scenario.

**Figure 15 sensors-23-01813-f015:**
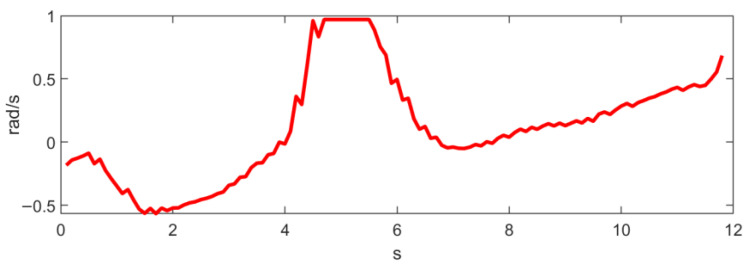
Angular velocity generated in the long wall scenario.

**Figure 16 sensors-23-01813-f016:**
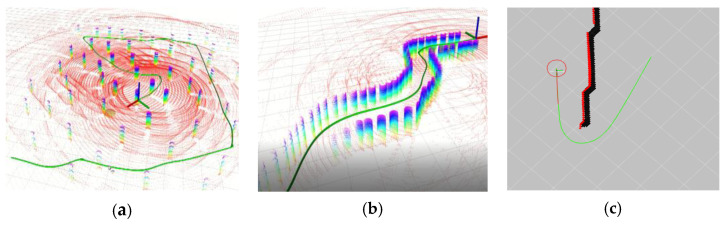
Planned path in (**a**) forest scenario; (**b**) narrow channel scenario; (**c**) long wall scenario. Here the 3D point clouds of the environment are depicted with different colors.

**Figure 17 sensors-23-01813-f017:**
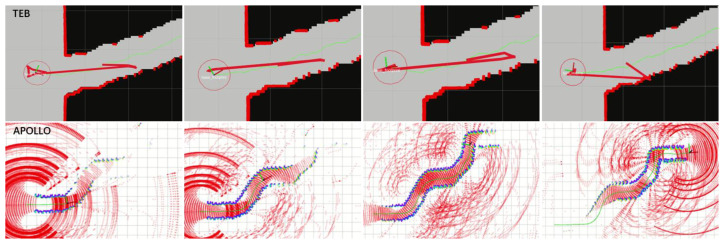
Illustration of the performance of the TEB algorithm and the APOLLO algorithm in the narrow channel scenario. For the TEB algorithm, the red line indicates the local trajectory and the green line indicates the global path. For the APOLLO algorithm, the blue dots represent the point cloud of the narrow channel.

**Figure 18 sensors-23-01813-f018:**
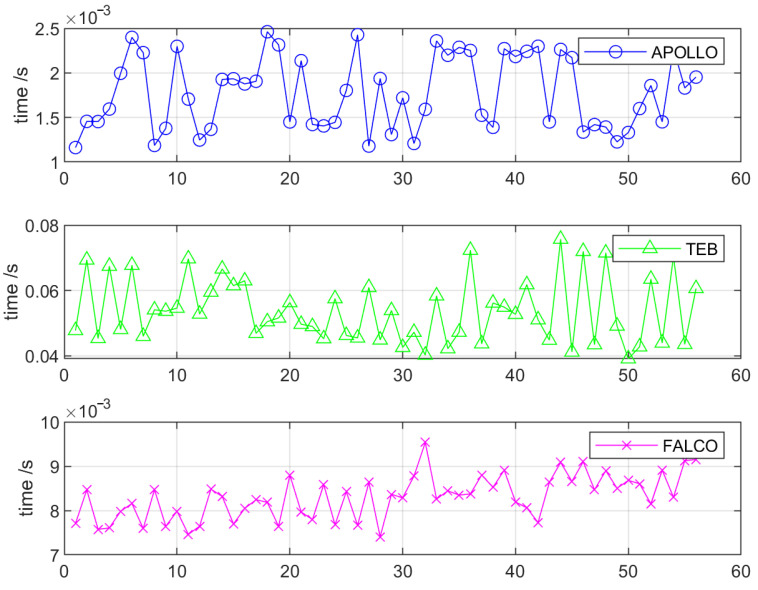
CPU time comparison of APOLLO, TEB and FALCO.

**Figure 19 sensors-23-01813-f019:**
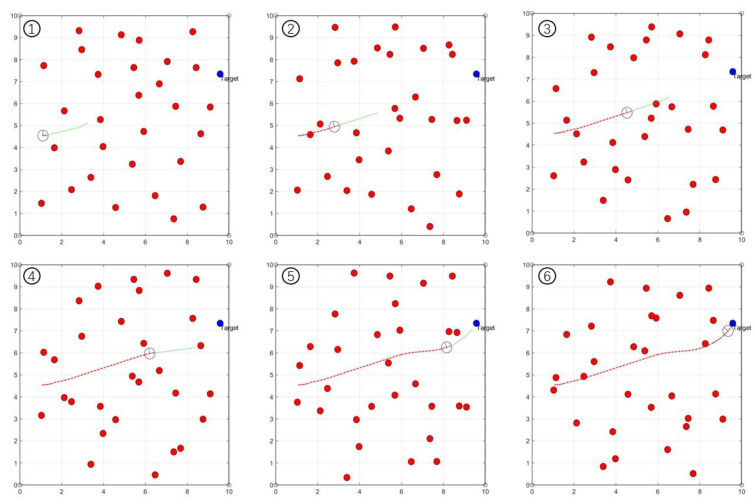
Trajectory generation process in the human-involved dynamic driving scenario. Red dots represent the randomly walking humans, and the hollow circle represents the vehicle. The number ①–⑥ indicate six sequential moments from the beginning to the end.

**Figure 20 sensors-23-01813-f020:**
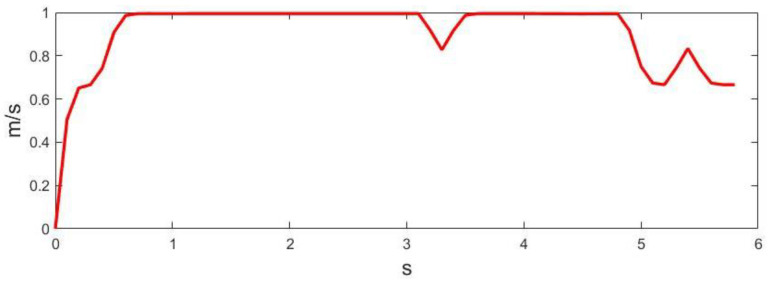
Linear velocity generated in the human-involved dynamic driving scenario.

**Figure 21 sensors-23-01813-f021:**
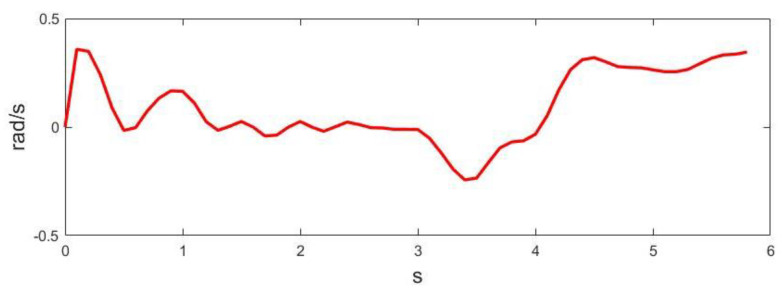
Angular velocity generated in the human-involved dynamic driving scenario.

## Data Availability

Not applicable.

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
