# Peer review of "Apollo: Adaptive Polar Lattice-Based Local Obstacle Avoidance and Motion Planning for Automated Vehicles"

_sensors, 2023, doi:10.3390/s23041813_

Round 1
Reviewer 1 Report
This paper mainly studies the following contents: path planning method based on adaptive polar coordinates, path optimization method based on the above methods and speed planning method. The overall structure is relatively smooth, the writing is fluent and the research content is novel.
However, this paper still has the following deficiencies:
1.The relationship between some variables is not well explained. For example, whether there is a certain relationship between and in formula (4).
2.The specific symbols are not well explained. For example, The relationship between,and in the path optimization algorithm, whether they are constants, etc.
3. In order to make the writing clear, It is recommended to mark on the picture (13);
4.Based on the experimental results shown in the picture (17), it is suggested to add a comparative experiment between the TEB algorithm and the algorithm in this paper.
5. Considering the speed planning algorithm proposed in this paper, it is recommended to add the corresponding speed planning diagram in the final simulation process (Figure 19).
Reviewer 2 Report
Highly interesting approach. Combined with neural 3D rendering in terms of environmental modeling, such an approach becomes really strong.
1. Please do explain how and when the offered and already simulated approach would be now tested in-situ, where, the method / missions (considering complex spatial situations).
2. Please emphasize also the vehicle swarms/lines etc. topic and how it exactly is relevant for what you offer by APOLLO and vice versa.
3. Please always clarify well how your work exactly would positively impact the daily life.
4. Please do explain the correlation and its potential variation effects when the amount of autonomous vehicles on the roads grows compared to the manned ones, and the failure/accident rate development then. Of course, under the use of APOLLO.
Round 2
